# COVID-19 Vaccine Hesitancy among Patients with Inflammatory Bowel Disease Receiving Biologic Therapies in Kuwait: A Cross-Sectional Study

**DOI:** 10.3390/vaccines10010055

**Published:** 2021-12-31

**Authors:** Mohammad Shehab, Yasmin Zurba, Ali Al Abdulsalam, Ahmad Alfadhli, Sara Elouali

**Affiliations:** 1Department of Internal Medicine, Mubarak Al-Kabeer University Hospital, Aljabreyah 46300, Kuwait; yasminzurba@moh.gov.kw (Y.Z.); aalabdulsalam@moh.gov.kw (A.A.A.); Ahalfadhli@moh.gov.kw (A.A.); 2Department of Internal Medicine, Cleveland Clinic, Abu Dhabi P.O. Box 112412, United Arab Emirates; elouals2@clevelandclinicabudhabi.ae

**Keywords:** COVID-19, vaccine, IBD, biologics, hesitancy, infliximab, vedolizumab

## Abstract

Background: COVID-19 vaccinations have been shown to be effective in reducing risk of severe infection, hospitalization, and death. They have also been shown to be safe and effective in patients with inflammatory bowel disease (IBD) who are receiving biologic therapies. In this study, we aimed to evaluate the prevalence of vaccination among patients receiving biologic therapies for IBD. Methods: A single-center prospective cross-sectional study conducted at a tertiary care inflammatory bowel disease center in Kuwait. Data from patients with inflammatory bowel disease (IBD) who attended the gastroenterology infusion clinic from 1 June 2021 until 31 October 2021 were retrieved. Patients who received infliximab or vedolizumab at least six weeks before recruitment were included. The primary outcome was prevalence of COVID-19 vaccination. The secondary outcome was to assess whether prevalence of COVID-19 vaccination differed based on sex, age, type of biologic therapy and nationality. Results: The total number of inflammatory bowel disease (IBD) patients enrolled in the study was 280 (56.0% male and 44.0% female). Of the total, 112 (40.0%) patients were diagnosed with ulcerative colitis and 168 (60.0%) with Crohn’s disease. The number of ulcerative colitis patients who were vaccinated was 49 (43.8%) and the number of Crohn’s disease patients who were vaccinated was 68 (40.5%). The median age was 33.2 years and BMI was 24.8 kg/m^2^. With respect to the total number of patients, 117 (41.8%) were vaccinated with either BNT162b2 or ChAdOx1 nCoV-19 and 163 (58.2%) were not vaccinated. Female patients were more likely to receive the vaccine compared to male patients (83.0% vs. 63.8%, *p* < 0.001). In addition, patients above the age 50 were more likely to receive the vaccine than patients below the age of 50 (95.6% vs. 31.2% *p* < 0.001). Expatriates were more likely to receive the vaccine than citizens (84.8% vs. 25.0%, *p* < 0.001). There was no statistical difference between patients on infliximab and vedolizumab with regard to prevalence of vaccination (40.0% vs 48.0%, *p* = 0.34). Conclusion: The overall prevalence of COVID-19 vaccination among patients with inflammatory bowel disease (IBD) on biologic therapies was lower than that of the general population and world health organization (WHO) recom-mendation. Female patients, patients above the age of 50, and expatriates were more likely to receive the vaccine. Physicians should reinforce the safety and efficacy of COVID-19 vaccines among patients, especially IBD patients on biologic therapies, who express hesitancy towards them.

## 1. Introduction

The global spread of Coronavirus disease 2019 (COVID-19) has impacted every aspect of life. It was declared a pandemic by the World Health Organization (WHO) on the 11 March 2020 [1]. Although the pandemic has not finished yet, the most effective way of ending it is with widespread vaccination [2]. Widespread coverage of vaccines may be hindered by vaccine hesitancy, defined by WHO as a “delay in acceptance or refusal of vaccination despite availability of vaccination services” [3]. Patients who suffer from immune-mediated inflammatory diseases, such as inflammatory bowel disease (IBD), are often treated with biologic therapies. Those patients may be hesitant to receive COVID-19 vaccines due to safety and efficacy concerns [4].

Interestingly, it has been shown in a recent meta-analysis that biologic therapies are not associated with severe COVID-19 or worse outcomes in patients with IBD [5]. In addition, COVID-19 vaccines have been shown to be effective in patients with IBD who are receiving biologic therapies [6,7]. The Surveillance Epidemiology of Coronavirus Under Research Exclusion (SECURE-IBD) is an international database that was established at the beginning of the COVID-19 pandemic to monitor and report the outcomes of COVID-19 infection in IBD patients. This registry includes the outcomes of more than 6000 IBD patients infected with COVID-19 from 72 countries [8]. The SECURE-IBD study showed that in patients with IBD and COVID-19 infection, biologic therapies are not associated with poor outcomes [9]. The International Organization for the study of Inflammatory Bowel Disease (IOIBD) recommends COVID-19 vaccine for patients with IBD [10].

In Kuwait, BNT162b2 (Pfizer/BioNTech, Mainz, Germany) and ChAdOx1 nCoV-19 (Oxford/AstraZeneca, Oxford, UK) vaccines are the only two available COVID-19 vaccines as of December 2020. The eligible age to receive BNT162b2 vaccine is 12 years or older, whereas for ChAdOx1 nCoV-19 it is 18 years or older. By the end of October 2021, the government of Kuwait announced that 74.0% of the eligible population had received at least one dose of either COVID-19 vaccine [11]. The success of a vaccination campaign is dependent on the acceptance of the general population, which can be influenced by the views of unvaccinated individuals. These groups, in turn, can pose a risk to vulnerable individuals, especially those living among them [12]. There are no previous studies on COVID-19 vaccine prevalence in patients with IBD in Kuwait. The aim of this study is to evaluate the prevalence of COVID-19 vaccines in patients with IBD receiving biologic therapies and to evaluate possible factors associated with vaccination hesitancy. Our study will contribute to identifying subgroups within the IBD population that should be targeted for COVID-19 vaccination campaigns, through which strategies can be developed to in-crease vaccination rates such as intensified education and addressing misconceptions regarding the vaccine.

## 2. Material and Methods

### Study Design and Recruitment

A single-center prospective cross-sectional study was conducted at Mubarak Al-Kabeer University Hospital, a tertiary care inflammatory bowel disease center. Data from patients with inflammatory bowel disease (IBD) who attended the gastroenterology infusion clinic from 1 June 2021 until 31 October 2021 were retrieved. During this period, the COVID-19 vaccine was available with easy access in all primary care centers in Kuwait free of charge. Vaccination counselling was provided by local health authorities and physicians during health visits. Patients were eligible to be included if they: (1) had a confirmed diagnosis of inflammatory bowel disease before the start of the study, (2) received infliximab or vedolizumab at least six weeks prior to recruitment; and (3) were 18 years of age or older. Exclusion criteria were the following: (1) prior suspected or confirmed severe acute respiratory syndrome Coronavirus 2 (SARS-CoV-2) infection, (2) corticosteroid use within two weeks of the recruitment date, (3) contraindication to receiving the COVID-19 vaccine as per the Food and Drug Administration (FDA) recommendations [13], (4) refusal to participate in the study. We avoided selection bias in the study design phase by restriction. We restricted the enrollment of study participants to those who were receiving biologic therapies in the infusion room, as opposed to other patients who receive biologic therapies through other modes of delivery elsewhere. Reliability was ensured by assessing the study participants’ vaccination status in two separate visits to the infusion room eight weeks apart.

The study was performed and reported in accordance with the Strengthening the Reporting of Observational Studies in Epidemiology (STROBE) guidelines (Appendix A). Patient and disease characteristics, demographics, and vaccination details were obtained from electronic medical records. Diagnosis of IBD was made according to the international classification of diseases (ICD-10 version: 2019). Patients were considered to have IBD when they had ICD-10 K50, K50.1, K50.8, K50.9 corresponding to Crohn’s disease (CD) or ICD-10 K51, K51.0, K51.2, K51.3, K51.5, K51.8, K51.9 corresponding to ulcerative colitis (UC) [14].

The primary outcome was the prevalence of COVID-19 vaccination among patients with IBD receiving infliximab or vedolizumab. The secondary outcome was assessment of whether the prevalence of COVID-19 vaccination differed based on sex, age, type of biologic therapy and nationality.

Analyses were conducted using R (R core team, 2017). The statistical significance level was set at α = 0.05. Descriptive analyses were conducted to calculate frequencies and proportions of categorical variables. χ2 tests were used to assess whether the prevalence of COVID-19 vaccination differed across categories of demographic variables. The magnitude of the effect was calculated for the comparisons, including effect size analysis, using the prevalence ratio (PR).

## 3. Results

The total number of inflammatory bowel disease (IBD) patients enrolled in the study was 280 (56.0% male and 44.0% female). The median age was 33.2 years, the median BMI was 24.8 kg/m^2^, and 58 (20.0%) of the participants were smokers. The most common comorbidities were asthma (13.6%), diabetes (6.7%), and arthritis (5.0%). The median duration of therapy at the time of the study was 12 months for patients on infliximab and 11 months for patients on vedolizumab. The participants included 112 patients with ulcerative colitis (UC) (40.0%) and 168 (60.0%) with Crohn’s disease (CD) (Table 1).

Among the study participants, 117 (41.8%) were vaccinated with either BNT162b2 or ChAdOx1 nCoV-19 and 163 (58.2%) were not vaccinated. With respect to UC and CD patients, 49 (43.8%) and 68 (40.5%) were vaccinated, whereas 63 (56.2%) and 100 (59.5%) were not, respectively. Of the vaccinated patients, 25 (21.4%) were vaccinated with one dose, whereas 92 (78.6%) were vaccinated with two doses. Of the total number of participants, 232 (82.9%) were receiving infliximab, and 48 (17.1%) were receiving vedolizumab. Ninety-four (40.0%) patients on infliximab were vaccinated, whereas 138 (60.0%) were not. On the other hand, 23 (48.0%) patients on vedolizumab were vaccinated and 25 (52.0%) were not. Regarding age, 46 (16.4%) participants were above the age of 50 (44 (95.6%) vaccinated, and 2 (4.4%) unvaccinated), whereas 234 (83.6%) were below the age of 50 (73 (31.2%) vaccinated, and 161 (68.8%) unvaccinated). Regarding nationality, 50 (25.0%) citizens were vaccinated and 151 (75.0%) were not. Among expatriates, 67 (84.8%) were vaccinated and 12 (15.2%) were not. Of the total number of male participants, 98 (62.4%) were vaccinated and 59 (37.6%) were not, and of the female participants, 102 (83.0%) were vaccinated and 21 (17.1%) were not. A total of five patients were pregnant, all of whom were not vaccinated (Table 2).

Female patients were 1.3 times more likely to receive the vaccine compared to male patients (83.0% vs. 63.8%, *p* < 0.001). In addition, participants above the age of 50 were 3.1 times more likely to receive the vaccine than participants below the age of 50 (95.6% vs. 31.2% *p* < 0.001). Expatriates were 3.4 times more likely to receive the vaccine than citizens (84.8% vs. 25.0%, *p* < 0.001) (Table 2, Figure 1). However, there were no statistical differences between patients on infliximab and vedolizumab in terms of prevalence of vaccination (40.0% vs. 48.0%, *p* = 0.34) (Table 2, Figure 2).

## 4. Discussion

The world’s leading inflammatory bowel disease (IBD) medical organizations recommend COVID-19 vaccination for IBD patients on biologic therapies; however, for unclear reasons, there remains an evident hesitancy to receiving the vaccine [15].

In this study, we report that among 280 IBD patients on intravenous biologic therapy, 117 (41.8%) were vaccinated and 163 (58.2%) were not vaccinated. This acceptance rate is less than that of the general population (74.0%) [11]. According to World Health Organization (WHO) experts, the target vaccination rate in the population of every country is 60–80% in order to achieve herd immunity and break the chain of transmission [16,17]. A cross-sectional study conducted in Saudi Arabia in October 2021 reported that the majority of IBD patients (68.0%) had received the COVID-19 vaccine, which is higher than the uptake found in our study [18]. In a cross-sectional study conducted in Italy, decreased knowledge about COVID-19 vaccines and vaccine hesitancy were found to be directly related. This emphasizes the importance of providing accurate information and intensifying public health campaigns in order to increase education and in turn COVID-19 vaccine acceptance [12]. 

In our study, positive predictors of willingness to receive COVID-19 vaccination were found to be female sex, older age (above 50 years old), and expatriate status. Negative predictors were male sex, younger age (below 50 years old), and citizens. Higher vaccination rates among females could be explained by the fact that they were found to have higher rates of medical services utilization, including healthcare services for prevention, detection, and treatment [19]. In addition, females make up the majority of frontline healthcare workers, putting them at higher occupational exposure [20]. A possible rationale for higher vaccination rate among patients above the age of 50 is the higher prevalence of comorbidities in this age group and fear of severe COVID-19 infection. This is consistent with the available data in the literature, as older age has been associated with higher COVID-19 vaccination rates [21]. Higher vaccination rate among expatriates may be due to the fact that they are more likely to travel to their countries of origin to visit their families, and local health regulations allow only vaccinated individuals to travel abroad. Moreover, most expatriates work in the private sector, which requires positive vaccination status in order to return to work. Notably, the percentage of vaccinated patients receiving infliximab and vedolizumab were similar, 40.0% and 48.0% respectively. A cohort study conducted in the United Kingdom (UK) also showed similar vaccine uptake among IBD patients treated with infliximab and vedolizumab [22]. A German case-control study concluded that IBD participants were significantly more hesitant to receive the COVID-19 vaccine (58.5%) compared to the control group (65.1%) [23].

Females of child-bearing age or in the peripartum period were found to have lower vaccination rates [22], which is in line with the results of our study, in which none of the pregnant participants were vaccinated. This may be due to fear of long-term effects on the mother or fetus despite that COVID-19 vaccines having been recommended by the Royal College of Obstetricians and Gynecologists (RCOG) and the American College of Obstetricians and Gynecologists (ACOG) [24,25]. A multi-method study on acceptance towards COVID-19 vaccination by females showed an 81.2% acceptance rate among non-pregnant females. This number dropped significantly to 62.1% in pregnant females, which was attributed to vaccine safety concerns [26]. Therefore, it is evident that specific efforts should be aimed towards females of childbearing age or in the peripartum period in order to address any misconceptions they may have regarding the COVID-19 vaccine, and to enhance their trust in it.

The International Organization for the study of Inflammatory Bowel Disease (IOIBD) is an organization of clinical researchers devoted to the study and management of IBD [27]. An international consensus meeting among members of the IOIBD took place to establish recommendations concerning the use of COVID-19 vaccines in patients with IBD. They concluded that all IBD patients, including patients on immune-modifying medications, should be vaccinated against COVID-19. Studies have reported that patients with IBD, including those on biologic therapy, do not have increased risk of flare-ups post-COVID-19 immunization [28]. Strong antibody responses were found in the participants who received any COVID-19 vaccine; therefore, patients with IBD are expected to develop immunity from any of the available vaccine strategies, regardless of treatment with immuno-modifying therapies [29]. Similar results were reported in two regional studies that examine the serological response of COVID-19 vaccines in IBD patients receiving biologic therapies [30,31]. This evidence addresses claims that patients on biologic therapies do not mount an immune response to COVID-19 vaccines and can be used to encourage the acceptance of the vaccines. The serological response of COVID-19 vaccines was not only studied on IBD patients on biologic therapies but also on cancer patients receiving immunotherapy and chemotherapy, who are also considered a vulnerable group. In recent studies, seroconversion after COVID-19 vaccination has been demonstrated in cancer patients treated with immune checkpoint inhibitors (ICI). Moreover, these patients were found to have similar survival outcomes to the general population when infected with COVID-19 [32,33,34,35,36].

Ever since the start of the pandemic, there has been a wave of misinformation, rumors, and conspiracy theories on social media platforms, such as the perception that COVID-19 vaccines alter human DNA, that a microchip is implanted in the person’s body when they are injected, or that COVID-19 vaccines are live vaccines that can cause more harm than benefit [37]. This has led to anti-vaccination movements worldwide. Such movements have been found to be associated with lower rates of vaccine acceptance during pandemics and disease outbreaks [38]. In the literature, study participants with IBD listed the fast development of COVID-19 vaccines, fear of worsening of their underlying dis-ease, and concerns about the COVID-19 vaccines’ adverse effects as the main reasons they were hesitant to receive it [4,15]. Other reasons for COVID-19 vaccine hesitancy included belief that a COVID-19 vaccine is not needed for protection as the nature of the disease is harmless, opposition to all vaccinations in general, lack of trust in the healthcare system, uncertainty regarding COVID-19 vaccine efficacy, and the assumption that prior COVID-19 infection leads to long-term immunity [39]. Health care authorities should develop strategies to tackle the spread of COVID-19 misinformation online. Examples of such strategies might include implementing educational and online social campaigns that provide trusted information, or designating a hotline to address COVID-19 vaccine concerns.

To determine whether these findings can be extended to the general patient population, larger studies in patients of diverse demographics are required. Moreover, the results of this study can be used to aid physicians, especially IBD specialists, researchers, and public health departments, to develop and apply interventions to overcome COVID-19 vaccine hesitancy.

Our study is a well-designed cross-sectional study targeting patients with IBD on biologic therapies. It evaluates factors that are associated with hesitancy to vaccinate in this vulnerable population. In addition, it is the first study in Kuwait to evaluate the prevalence of COVID-19 vaccination among IBD patients. A relatively small sample size is a limitation of our study. Given that this is a cross-sectional observational study, selection bias is possible. Therefore, it was avoided in the study design by restricting the study participants to those who were receiving biologic therapies in the infusion room. Participants who had been infected with COVID-19 were excluded from our study. In the future, it would be interesting to assess COVID-19 disease severity in IBD patients receiving biologic therapies in relation to their vaccination status. In addition, the prevalence of vaccination in patients with IBD on other biologic therapies (such as adalimumab and ustekinumab) was not evaluated.

## 5. Conclusions

The overall prevalence of COVID-19 vaccination among patients with inflammatory bowel disease (IBD) on biologic therapies was low-er than that of the general population and world health organization (WHO) recommendations. Female patients, patients above the age of 50, and expatriates were more likely to receive the vaccine. Physicians should rein-force the safety and efficacy of COVID-19 vaccines among patients, especially IBD patients on biologic therapies who express hesitancy towards them. Future studies using larger and diverse patient populations are required to generalize the results more widely.

## Figures and Tables

**Figure 1 vaccines-10-00055-f001:**
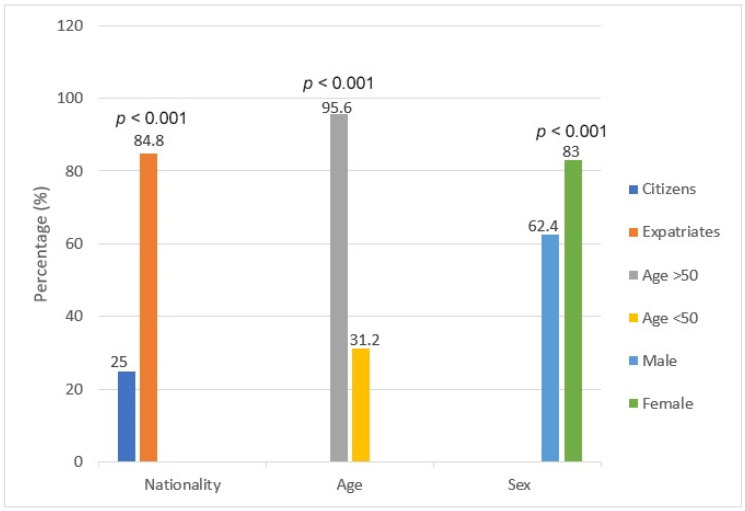
Demographics of patients with IBD based on vaccination status.

**Figure 2 vaccines-10-00055-f002:**
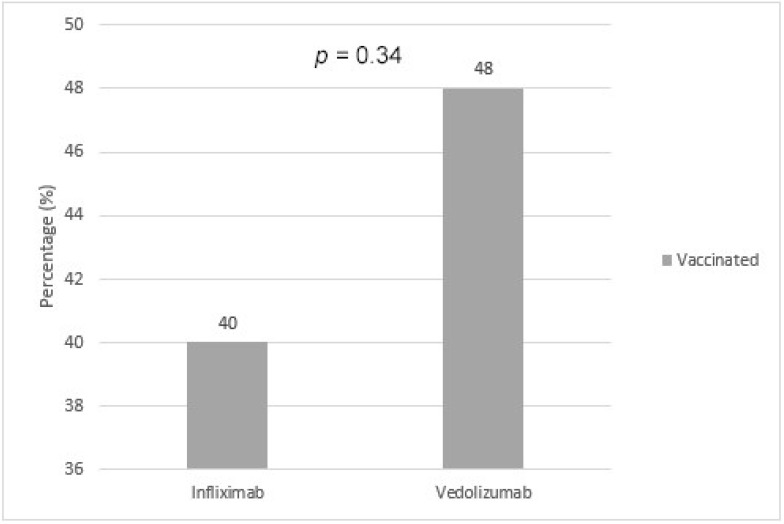
Vaccination status of patients with IBD based on type of biologic therapy.

**Table 1 vaccines-10-00055-t001:** Demographics of Patients with IBD.

Variable	Study Group (n = 280)
Mean age (years)	33.2
Sex n (%)	
Male	157 (56.0%)
Female	123 (44.0%)
BMI (Median)	24.8
Smoking n (%)	58 (20.0%)
Comorbidities n (%)	
Diabetes	19 (6.7%)
OSA	5 (1.7%)
Hypertension	9 (3.2%)
Cardiovascular Disease	9 (3.2%)
Arthritis	14 (5.0%)
Kidney	9 (3.2%)
Asthma	38 (13.6%)
Hyperlipidemia	9 (3.2%)
Median infliximab therapy (months) Median vedolizumab therapy (months)	12 11
Disease extent, n (%)	
Ulcerative colitis (UC)	112 (40.0%)
E1: ulcerative proctitis	20 (17.8%)
E2: left sided colitis	32 (28.5%)
E3: extensive colitis	60 (53.6%)
Crohn’s disease (CD)	168 (60.0%)
L1: ileal	84 (50.0%)
L2: colonic	20 (11.9%)
L3: ileocolonic	59 (35.2%)
L4: upper gastrointestinal	5 (2.8%)
B1: inflammatory	75 (44.6%)
B2: stricturing	44 (26.2%)
B3: penetrating	49 (29.2%)

**Table 2 vaccines-10-00055-t002:** Demographics of patients with IBD according to vaccination status.

Demographics/Vaccination Status	Total Number of Patients (280)	Vaccinated Patients (117)	Unvaccinated Patients (163)
Ulcerative Colitis	112	49 (43.8%)	63 (56.2%)
Crohn’s Disease	168	68 (40.5%)	100 (59.5%)
One dose	25	25	N/A
Two doses	92	92	N/A
Infliximab	232	94 (40.0%)	138 (60.0%)
Vedolizumab	48	23 (48.0%)	25 (52.0%)
Age > 50	46	44 (95.6%)	2 (4.4%)
Age < 50	234	73 (31.2%)	161 (68.8%)
Citizens	201	50 (25.0%)	151 (75.0%)
Expatriates	79	67 (84.8%)	12 (15.2%)
Male	157	98 (62.4%)	59 (37.6%)
Female	123	102 (83.0%)	21 (17.1%)
Pregnant	5	0	5

## Data Availability

The data presented in this study are available on request from the corresponding author. The data are not publicly available due to legal and ethical restrictions.

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
