# Peer review of "COVID-19 Vaccine Hesitancy among Patients with Inflammatory Bowel Disease Receiving Biologic Therapies in Kuwait: A Cross-Sectional Study"

_vaccines, 2021, doi:10.3390/vaccines10010055_

Round 1

Reviewer 1 Report

First of all, I would like to thank for the opportunity to review this paper. Actually, the vaccination campaign is the first method to counteract the COVID-19 pandemic. However, sufficient vaccination coverage is conditioned by the people’s acceptance of these vaccines, especially in at risk groups of population, such as those affected by inflammatory bowel disease (IBD). In this context, the paper under evaluation is aimed at evaluate the prevalence of COVID-19 vaccines in patients with IBD on biologic therapies and evaluate possible factors that are associated with vaccination.

The article is interesting, but it must be improved in order to be suitable for publication in an international journal.

Title: it is overstated since it is carried on in one center enrolling a small sample of patients, I suggest to better identify the place where it was performed and that is a sample of patients.

Introduction: The authors should better clear what is the gap in the literature that is filled with this study. Moreover, since the vaccination campaign success is influenced by people’s acceptance of these vaccines, this issue must be discussed in the introduction reporting level of acceptance also in the general population (refer to Gallè, F. et al Knowledge and Acceptance of COVID-19 Vaccination among Undergraduate Students from Central and Southern Italy. Vaccines 2021, 9, 638). Finally, the Authors must explicit what is the potential contribution of the study to the literature and its implications.

Methods: The enrolment procedure must be better specified, is seems a little confusing who was involved in the survey? How did the authors choose the way used to enroll their sample? How did they avoid the selection bias? What is the reference population? what is the minimum sample related to the reference population and the power of the study? About the investigated factors, how did the authors choose what to associate? no mention to a validation process of the selected variable is reported. What about face validity, reliability and intelligibility?

Statistical analysis: I suggest to insert a measure of the magnitude of the effect for the comparisons, include an effect sizes analysis.

Discussion: It is sometimes redundant, it should be reorganized emphasizing the contribution of the study to the literature, and overall the limits. The authors report the results but it is not clear their practical impact. Results of the acceptance should be compared to those of general population (refer to Gallè, F. et al Knowledge and Acceptance of COVID-19 Vaccination among Undergraduate Students from Central and Southern Italy. Vaccines 2021, 9, 638). English should be improved.

Conclusion: In this paragraph the Authors report only results, but the readers want to know what is the message of the paper.

Author Response

Reviewer 1:
First of all, I would like to thank for the opportunity to review this paper. Actually, the
vaccination campaign is the first method to counteract the COVID-19 pandemic. However,
sufficient vaccination coverage is conditioned by the people’s acceptance of these vaccines,
especially in at risk groups of population, such as those affected by inflammatory bowel
disease (IBD). In this context, the paper under evaluation is aimed at evaluate the prevalence
of COVID-19 vaccines in patients with IBD on biologic therapies and evaluate possible
factors that are associated with vaccination.

The article is interesting, but it must be improved in order to be suitable for publication in
an international journal.

Title: it is overstated since it is carried on in one center enrolling a small sample of patients, I
suggest to better identify the place where it was performed and that is a sample of patients.

- Thank you, the title has been changed.

COVID-19 Vaccine Hesitancy Among Patients with Inflammatory Bowel Disease on
Biologic Therapies in Kuwait; A cross-sectional study

Introduction: The authors should better clear what is the gap in the literature that is filled
with this study. Moreover, since the vaccination campaign success is influenced by people’s
acceptance of these vaccines, this issue must be discussed in the introduction reporting
level of acceptance also in the general population (refer to Gallè, F. et al Knowledge and
Acceptance of COVID-19 Vaccination among Undergraduate Students from Central and
Southern Italy. Vaccines 2021, 9, 638). Finally, the Authors must explicit what is the potential
contribution of the study to the literature and its implications.

- Thank you, the gap in the literature that is filled with this study has been added to
the introduction: (
There are no previous studies on COVID-19 vaccine prevalence in
patients with IBD in Kuwait.
)
- Galle F. et el reference has been added:
The success of a vaccination campaign is
dependent on the acceptance of the general population, which can be influenced by
views of unvaccinated individuals. These groups in turn, pose a risk on vulnerable
individuals especially those living among them

- The level of acceptance in the general population is already mentioned in the
introduction:
By the end of October 2021, the government of Kuwait announced that
74.0% of the eligible population in Kuwait had received at least one dose of either
COVID-19 vaccine

- The contribution of our study to the literature has been added to the introduction:
Our study will contribute to identifying subgroups withing the IBD population that

should be targeted for COVID-19 vaccination campaigns and develop strategies to
increase vaccination rate such as addressing misconceptions about the vaccine

Methods: The enrolment procedure must be better specified, is seems a little confusing who
was involved in the survey? How did the authors choose the way used to enroll their
sample? How did they avoid the selection bias? What is the reference population? what is
the minimum sample related to the reference population and the power of the study?
About the investigated factors, how did the authors choose what to associate? no mention
to a validation process of the selected variable is reported. What about face validity,
reliability and intelligibility?

- All patients who attended the infusion clinic at
Mubarak Al-Kaber University Hospital,
a tertiary care inflammatory bowel disease center, from June 1st, 2021 until October
31st, 2021
who accepted to participate were included in the study.
- The exclusion criteria were: (see methods)

- We avoided selection bias in the study design phase by restriction. We restricted the
enrollment of study participants to those who are receiving biologic therapies only in
the infusion room as opposed to other patients who receive biologic therapies
through other modes of delivery elsewhere. However, given that this is a cross-
sectional observational study, selection bias is possible. Please see limitations
paragraph.

- Minimal sample to the reference population: this is a cross-sectional observational
study. All patients were included, and there was no comparison between two groups,
so no power or sample size need to be calculated.

- Regarding what we chose to associate, we selected common variables mentioned in
the literature such as age, sex, and nationality that we thought may be a factor.
Examples:

-
https://academic.oup.com/ibdjournal/advance-
article/doi/10.1093/ibd/izab172/6321213

- https://gut.bmj.com/content/70/10/1884

- Face validity, Reliability and intelligibility, although don’t apply to this cross-sectional
study,
reliability was ensured by assessing the study participants’ vaccination status in
two separate visits to the infusion room 8 weeks apart. See added method section.

Statistical analysis: I suggest to insert a measure of the magnitude of the effect for the
comparisons, include an effect sizes analysis.

- Thank you, the magnitude of the effect was calculated for the comparisons, including
effect size analysis, by the prevalence ratio (PR). Please see edited methods and results.

Discussion: It is sometimes redundant, it should be reorganized emphasizing the
contribution of the study to the literature, and overall the limits. The authors report the
results but it is not clear their practical impact. Results of the acceptance should be
compared to those of general population (refer to Gallè, F. et al Knowledge and Acceptance

of COVID-19 Vaccination among Undergraduate Students from Central and Southern Italy.
Vaccines 2021, 9, 638). English should be improved.

- Thank you, the discussion was edited and redundancies were omitted. Please see
edited version.

- Native English readers reviewed the manuscript twice.

- The contribution of study to the literature and practical impact was mentioned in the
discussion section of the edited manuscript.

- Results of the vaccine acceptance in our study were compared to the level of
acceptance in the general population, and the reference was used.

(In this study, we report that among 280 IBD patients on intravenous biologic
therapy, 117 (41.8%) were vaccinated and 163 (58.2%) were not vaccinated. This
acceptance rate is less than that of the general population (74.0%)).

(In a cross-sectional study conducted in Italy, decreased knowledge about COVID-19
vaccines and vaccine hesitancy were found to be directly related. This emphasizes the
importance of providing accurate information and intensifying public health
campaigns in order to increase education and in turn COVID-19 vaccine acceptance)

Conclusion: In this paragraph the Authors report only results, but the readers want to know
what is the message of the paper.

- Thank you, please see edited conclusion.

Reviewer 2 Report

Shehab and colleagues proposed an interesting article aimed at evaluating the adherence to COVID-19 vaccination in patients with inflammatory bowel diseases treated with monoclonal antibodies. Overall, the manuscript is interesting, however, there are some issues that the authors have to address before publication. Please see the minor/major comments reported below:
1) The entire manuscript should be revised by an English native speaker;
2) How many patients with Chron’s disease and how many patients with ulcerative colitis were vaccinated and non-vaccinated? Please, clarify;
3) It would be interesting to describe also how many non-vaccinated patients treated with infliximab and vedolizumab and how many vaccinated patients treated with the same drugs got COVID-19 and develop severe symptoms. Please add these important data;
4) Please avoid redundancies in the Discuss section (e.g. the following sentence is redundant: “Females were more willing to receive the COVID-19 vaccine in relation to males.”);
5) The Discussion section is too verbose and contains no useful information. Consider to significantly shorten this section;
6) The authors should discuss the adherence to COVID-19 vaccination as well as the severity of infection in other diseases treated with monoclonal antibodies like tumors treated with immune checkpoint inhibitors. For this purpose, please see:
- PMID: 34147290
- PMID: 33491759
- PMID: 34830983
- PMID: 34601285
- PMID: 32785162

Author Response

Reviewer 2:
Shehab and colleagues proposed an interesting article aimed at evaluating the adherence to
COVID-19 vaccination in patients with inflammatory bowel diseases treated with
monoclonal antibodies. Overall, the manuscript is interesting, however, there are some
issues that the authors have to address before publication. Please see the minor/major
comments reported below:

1) The entire manuscript should be revised by an English native speaker;

Thank you, the edited manuscript has been revised by a native English speaker twice.

2) How many patients with Crohn’s disease and how many patients with ulcerative colitis
were vaccinated and non-vaccinated? Please, clarify;

Thank you, Added to Table 2

3) It would be interesting to describe also how many non-vaccinated patients treated with
infliximab and vedolizumab and how many vaccinated patients treated with the same drugs
got COVID-19 and develop severe symptoms. Please add these important data;

This is a very interesting questions, however, this was not the scope of our study. We will
consider it in a follow up study.

4) Please avoid redundancies in the Discuss section (e.g. the following sentence is
redundant: “Females were more willing to receive the COVID-19 vaccine in relation to
males.”);

Redundancies have been omitted.

5) The Discussion section is too verbose and contains no useful information. Consider to
significantly shorten this section;

The discussion section has been shortened and made more concise.

6) The authors should discuss the adherence to COVID-19 vaccination as well as the severity
of infection in other diseases treated with monoclonal antibodies like tumors treated with
immune checkpoint inhibitors. For this purpose, please see:

- PMID: 34147290

- PMID: 33491759

- PMID: 34830983

- PMID: 34601285

- PMID: 32785162

- Thank you. These studies cover a very interesting subject.

Reviewer 3 Report

This is a straightforward cross sectional study of patients in an inflammatory bowel disease clinic in Kuwait..

Introduction

There is no formal literature search so that we know which databases were searched and in which languages. This would help readers wishing to update the search later.

Sample: There is no discussion of refusals.

Methods. There is no discussion of how or where patients acquired the vaccine, and what counselling they reviewed. Importantly vaccine hesitancy (the title of this article) was not assessed in light of their concerns about their bowel disease and immune status and counselling they received or did not receive about these topics.

82.9% received infliximab and 78.6% two vaccine doses. What conclusions can be drawn about the 17.1 % who received vedoluziumab and the 21.4% one dose of vaccine?

The article for a cross sectional study is overly long and could be with value reduced in length by 15%. 

Author Response

Reviewer 3:
This is a straightforward cross sectional study of patients in an inflammatory bowel disease
clinic in Kuwait..

Introduction

There is no formal literature search so that we know which databases were searched and in
which languages. This would help readers wishing to update the search later.

- Thank you. This is a cross-sectional study, not systematic or narrative review, so
search database not usually added. We searched the literature in the English
language through Medline database.

Sample: There is no discussion of refusals.

- Study subjects refusing to participate in the study were excluded. Please see edited
exclusion criteria in the methods section.

Methods. There is no discussion of how or where patients acquired the vaccine, and what
counselling they reviewed. Importantly vaccine hesitancy (the title of this article) was not
assessed in light of their concerns about their bowel disease and immune status and
counselling they received or did not receive about these topics.

- The vaccine is available in all primary care centres in Kuwait. Vaccine counselling was
provided by local health authorities and physicians during clinic visits. Please see
added method section.

- Concerns of worsening of immune status with vaccination was discussed in the
discussion.

(In the literature, study participants with IBD listed the fast development of COVID-19
vaccine, fear of worsening of their underlying disease, and concern about COVID-19
vaccine adverse effects as the main reasons they were hesitant to receive it)

82.9% received infliximab and 78.6% two vaccine doses. What conclusions can be drawn
about the 17.1 % who received vedolizumab and the 21.4% one dose of vaccine?

- Of the study participants, 17.1% received vedolizumab, which is a less common type
of biologic therapy used to treat IBD compared to infliximab.

- 21.4% of the patients vaccinated with one dose had appointments scheduled for
their second dose at a date beyond the study period.

The article for a cross sectional study is overly long and could be with value reduced in
length by 15%.

- Thank you, the discussion section was reduced accordingly. Please see the edited
version.

Round 2

Reviewer 1 Report

The paper was improved according to the comments provided and it is now suitable for pubblication

Reviewer 2 Report

The authors partially answered my previous comments. In particular, they did not properly address my previous comments No. 3 and No. 6. 

Previous comment 3): Since the enrollment of patients was closed on 31 October, the authors could check the medical records to obtain the data on potential COVID-19 infection during the administration of the monoclonal antibodies. This further information will significantly improve the contents of the study paving the scientific basis to encourage the vaccination of patients who are treated with mAbs;

Previous comment 6): The authors just answered "Thank you. These studies cover a very interesting subject." without adding any information in the manuscript. Please properly address this point.

Author Response

Reviewer 2:
The authors partially answered my previous comments. In particular, they did not properly address my previous comments No. 3 and No. 6.

Previous comment 3): Since the enrollment of patients was closed on 31 October, the authors could check the medical records to obtain the data on potential COVID-19 infection during the administration of the monoclonal antibodies. This further information will significantly improve the contents of the study paving the scientific basis to encourage the vaccination of patients who
are treated with mAbs;

Thank you. It would indeed be an interesting addition, however, we do not have this data as it was not the primary outcome of our study. Patients who had been infected with COVID-19 were excluded from our study as mentioned in the methods section. As this is an interesting point and would make for good comparisons, we have added it to our discussion:
(Participants who had been infected with COVID-19 were excluded from our study. For the future, it would be interesting to assess COVID-19 disease severity in IBD patients on biologic therapies in relation to their vaccination status.)

Previous comment 6): The authors just answered "Thank you. These studies cover a very interesting subject." without adding any information in the manuscript. Please properly address this point.

Thank you. We have added information regarding this topic in the discussion and sited all the interesting articles provided to us.
(The serological response of COVID-19 vaccines was not only studied on IBD patients on biologic therapies but also on cancer patients receiving immunotherapy and chemotherapy, who are also considered a vulnerable group. In recent studies, seroconversion after COVID-19 vaccination has been demonstrated in cancer patients treated with immune checkpoint inhibitors (ICI). Moreover, these patients were found to have similar survival outcomes to the general population when infected with COVID-19.)

Reviewer 3 Report

Thanks to the authors , who have responded to the reviewers' suggestions
